METHODS AND RESOURCES

# Proteome-wide identification of HSP70/ HSC70 chaperone clients in human cells

**Seung W. Ryu**[1], **Rose Stewart**[1], **D. Chase Pectol**[2], **Nicolette A. Ender**[1], **Oshadi Wimalarathne**[1], **Ji-Hoon Lee**[1], **Carlos P. Zanini**[3], **Antony Harvey**[4], **Jon M. Huibregtse**[1], **Peter Mueller**[3], **Tanya T. Paull**[1]*

1 The Department of Molecular Biosciences, The University of Texas at Austin, Austin, Texas, United States of America, 2 The Department of Chemistry, Texas A&M University, College Station, Texas, United States of America, 3 Department of Statistics & Data Sciences, University of Texas at Austin, Austin, Texas, United States of America, 4 Thermo Fisher Scientific, Austin, Texas, United States of America

* tpaull@utexas.edu

**Data Availability Statement:** All relevant data are within the paper and its Supporting Information files.

**Funding:** The Howard Hughes Medical Institute contributed to the funding of this work. The

## Abstract

The 70 kDa heat shock protein (HSP70) family of chaperones are the front line of protection from stress-induced misfolding and aggregation of polypeptides in most organisms and are responsible for promoting the stability, folding, and degradation of clients to maintain cellular protein homeostasis. Here, we demonstrate quantitative identification of HSP70 and 71 kDa heat shock cognate (HSC70) clients using a ubiquitin-mediated proximity tagging strategy and show that, despite their high degree of similarity, these enzymes have largely nonoverlapping specificities. Both proteins show a preference for association with newly synthesized polypeptides, but each responds differently to changes in the stoichiometry of proteins in obligate multi-subunit complexes. In addition, expression of an amyotrophic lateral sclerosis (ALS)-associated superoxide dismutase 1 (SOD1) mutant protein induces changes in HSP70 and HSC70 client association and aggregation toward polypeptides with predicted disorder, indicating that there are global effects from a single misfolded protein that extend to many clients within chaperone networks. Together these findings show that the ubiquitin-activated interaction trap (UBAIT) fusion system can efficiently isolate the complex interactome of HSP chaperone family proteins under normal and stress conditions.

## Introduction

Every cell has a finely tuned balance of protein expression, folding, complex formation, localization, and degradation that is specific to its growth state and environmental cues. Many pathological states involve changes in this balance, resulting in loss of regulatory control or loss of resilience during stress [1,2]. While gene expression is now quantitatively measured at the level of nucleic acids and polypeptides, determining the status of cells with respect to protein homeostasis is still a challenge, as we lack the tools to examine folding of individual proteins and the functionality of multicomponent complexes quantitatively on a global scale.

The front line of defense against challenges to protein homeostasis is composed of cellular chaperones—proteins that recognize unfolded, aggregated, or unstable polypeptides [3]. These

funders had no role in study design, data collection and analysis, decision to publish, or preparation of the manuscript.

**Competing interests:** The authors have declared that no competing interests exist.

**Abbreviations:** AA, amino acid; ABC, ammonium bicarbonate; ALS, amyotrophic lateral sclerosis; ANXA1, annexin A1; AP-MS, affinity purification coupled with mass spectrometry; ATP1A1, ATPase, Na+/K+ transporting subunit alpha 1; ATP1B1, ATPase, Na+/K+ transporting subunit beta 1; BAG, Bcl-2–associated athanogene domain protein; BirA, biotin ligase; CCT, chaperonin-containing tailless complex; CLIPS, chaperones linked to protein synthesis; CMA, chaperone-mediated autophagy; DMEM, Dulbecco's Modified Eagle Medium; DPYSL2, dihydropyrimidinase like 2; ER, endoplasmic reticulum; FDR, false discovery rate; GAPDH, glyceraldehyde 3-phosphate dehydrogenase; GFP, green fluorescent protein; GRP, glucose-regulated protein; HIP, Hsp70-interacting protein; HOP, Hsp70-Hsp90 organizing protein; HSC, heat shock cognate; HSP, heat shock protein; LC-MS/MS, liquid chromatography–mass spectrometry; Mre11, meiotic recombination 11 protein; MSH, MutS homolog; OE, overexpressed; P bodies, processing bodies; PLIN3, perilipin 3; SGTA, small glutamine rich tetratricopeptide repeat containing alpha protein; shRNA, small hairpin RNA; SILAC, stable isotope labeling with amino acids in cell culture; SOD1, superoxide dismutase 1; SSA, stress-seventy subfamily A; SSB, stress-seventy subfamily B; TCP, T-complex protein; TRiC, TCP1-ring complex; UBAIT, ubiquitin-activated interaction trap; ΔGG, lacking the C-terminal Gly-Gly residues of ubiquitin.

ATP-dependent enzymes serve as a buffer to ensure that unfolded or aggregated polypeptides are either degraded or shielded within chaperone complexes or large protein-RNA assemblies [4]. The heat shock protein (HSP70) family of chaperones is conserved in all organisms and, together with HSP90 chaperones and a host of co-chaperones including the HSP40 family, is thought to be responsible for recognition of nascent unfolded proteins and protein aggregates [5–7]. In addition, HSP70 enzymes respond to heat and other forms of stress that result in protein destabilization and also contribute to protein degradation [8]. Mammals have two primary classes of HSP70: heat shock cognate (HSC70), which is constitutively expressed, and HSP70, which is present in normally growing cells but is also dramatically induced during heat exposure, other forms of stress, or depletion of HSC70 [9]. In addition, organelle-specific forms of HSP70 also serve specialized functions such as the mitochondria-specific glucose-regulated protein 75 (GRP75) and endoplasmic reticulum (ER)-specific 78 kDa glucose-regulated protein (GRP78) [10].

HSC70 and HSP70 are 85% identical to each other and are thought to have similar cellular roles [9]. One of the most important of these is to contribute to the folding of nascent polypeptides, and both HSC70 and HSP70 have been reported to associate with translating ribosomes [11,12]. In yeast, the stress-seventy subfamily A (SSA) family of HSP70 chaperones are grouped functionally with "chaperones linked to protein synthesis" (CLIPS) due to polysome association and similarities in transcriptional regulation in response to stress [13]. In addition, recent evidence suggests that HSP70 chaperones are essential for ongoing translation in mammalian cells, as the cessation of translation in response to heat is due to loss of the chaperones from actively translating ribosomes [14]. HSC70/HSP70 also safeguard proteins against aggregation, localize aggregated or misfolded proteins to stress granules or processing bodies (P bodies), promote correct trafficking of membrane-associated factors, and in some cases promote degradation via the proteasome or autophagy pathways [6–8].

While HSC70 (expressed from the *HSPA8* gene) and HSP70 (expressed from the *HSPA1A/B* genes) chaperones are generally considered to be redundant, there is substantial evidence supporting a division of functional roles. Deletion of *HSPA1A* and *HSPA1B* genes in mouse models generates sensitivity to stress and genomic instability but otherwise viable and fertile animals, whereas deletion of *HSPA8* is lethal in vertebrates [9,15]. The *HSPA8* gene from humans is able to complement yeast cells deficient in all cytoplasmic *SSA1-4* HSP70 genes, whereas *HSPA1A* cannot [16], suggesting that the properties of these two chaperones are not identical and that their client specificities may also be distinct.

Some of the clients (targets) of HSP70 chaperones have been determined empirically by investigation of specific proteins and their binding partners, but this has not been investigated on a global scale due to the transient nature of many chaperone-client interactions and the presumably large number of cellular clients [17]. Recent work identified the interactome of the ribosome-specific stress-seventy subfamily B (SSB1 and SSB2) chaperones in budding yeast using ribosome profiling, showing that these proteins associate with approximately 80% of nascent polypeptides [18]; however, the clients of cytosolic HSP70 proteins have not been identified comprehensively in yeast or mammalian cells. There have been several studies examining the interactomes of chaperones using affinity purification coupled with mass spectrometry (AP-MS) and meta-analysis of physical and functional interactions in budding yeast that have illuminated the extensive associations between HSP90 complexes, HSP70 complexes, associated co-chaperones, proteasome components, and other cofactors in eukaryotes [19–24]. These have primarily been focused on HSP90 interactors, but some clients of HSP70 have been identified, particularly in budding yeast [19–21]. Although AP-MS is a powerful technique, it is limited by the fact that only interactions at the time of collection are able to be identified, biasing results towards the most abundant targets. In addition, large protein complexes

may be isolated when only one or a few components of the complex actually interact with the chaperone.

Here, we use a ubiquitylation-based strategy [25] to covalently link chaperones to their binding partners, and identify the clients and cofactors of constitutive HSC70 and heat-inducible HSP70 in human cells. We find that HSP70 and HSC70 are largely nonoverlapping for client association under normal growth conditions, although both chaperones show a bias toward binding to newly synthesized polypeptides. Consistent with this role, HSC70 preferentially associates with proteins that are normally found in multicomponent complexes but lack their partners—proteins which are normally degraded. Lastly, expression of low levels of a misfolded protein drastically changes the client association landscape for HSC70 toward proteins with intrinsic disorder, suggesting that this chaperone is poised to alter its profile in response to subtle changes in the concentrations of misfolded proteins in cells.

## Results

### Identification of UBAIT clients

To identify chaperone binding partners, we employed a recently developed technique for the identification of protein–protein interactions that utilizes ubiquitin fusion proteins to covalently trap a protein of interest to its binding partners [25] (ubiquitin-activated interaction trap ["UBAIT"], Fig 1A). Fusion of ubiquitin to the C terminus of a bait protein generates a UBAIT protein, which can be charged by the ubiquitin E1 enzyme and transferred to an E2 protein. The E2-charged UBAIT can react *in trans* with lysine residues on proteins that interact with the bait protein, resulting in a stable amide linkage between the bait and interactors. The ubiquitin moiety of the UBAIT carries a K48R mutation to block potential K48 chain formation that can lead to proteasome degradation of the UBAIT. One of the primary advantages of this system is that interacting proteins accumulate as covalent ubiquitination conjugates to the bait protein, so that even transient binding partners can be readily identified by mass spectrometry. We tagged human HSC70 and HSP70 chaperones with ubiquitin at the C terminus and a biotinylation signal for *Escherichia coli* biotin ligase (BirA) and a V5 epitope tag at the N terminus. The biotin tag allows us to carry out the isolation of UBAIT conjugates under denaturing conditions, eliminating the presence of proteins that are associated with the target protein in non-covalent complexes. An important control is to express the UBAIT with deletion of the C-terminal Gly-Gly residues of the ubiquitin moiety ("ΔGG"), which prevents conjugation of the UBAIT. Inducible expression of the wild-type HSP70 and HSC70 UBAITs in human cells generated a ladder of larger products, consistent with covalent trapping of the UBAIT with client proteins or co-chaperones (Fig 1B); the ΔGG UBAITs did not generate conjugates, as expected. Levels of the UBAIT fusions were comparable to endogenous chaperones (S1 Fig). We know that an HSP70 enzyme with a C-terminal ubiquitin fusion is functional, based on experiments in a *Saccharomyces cerevisiae* strain lacking all four HSP70 orthologs (*SSA1-4*). In these experiments, expression of a yeast Ssa1 UBAIT ΔGG fusion fully complemented a *Δssa1-4* deleted strain for viability (S2 Fig).

To identify high-confidence chaperone binding partners of human HSC70 and HSP70, we used human osteosarcoma (U2OS) cell lines containing stably integrated, doxycycline-inducible HSC70-UBAIT (wild-type and ΔGG) and HSP70-UBAIT (wild-type and ΔGG) chaperones and grew these cells for 3 days after chaperone induction. We performed 12 wild-type UBAIT isolations and an equal number of control ΔGG isolations, each starting with 3 mg of total cellular protein, which were all performed under denaturing conditions followed by label-free quantification of the isolated material, as well as the total cell lysates by liquid chromatography–mass spectrometry (LC-MS/MS) (Fig 1C) [26]. We compared each of the

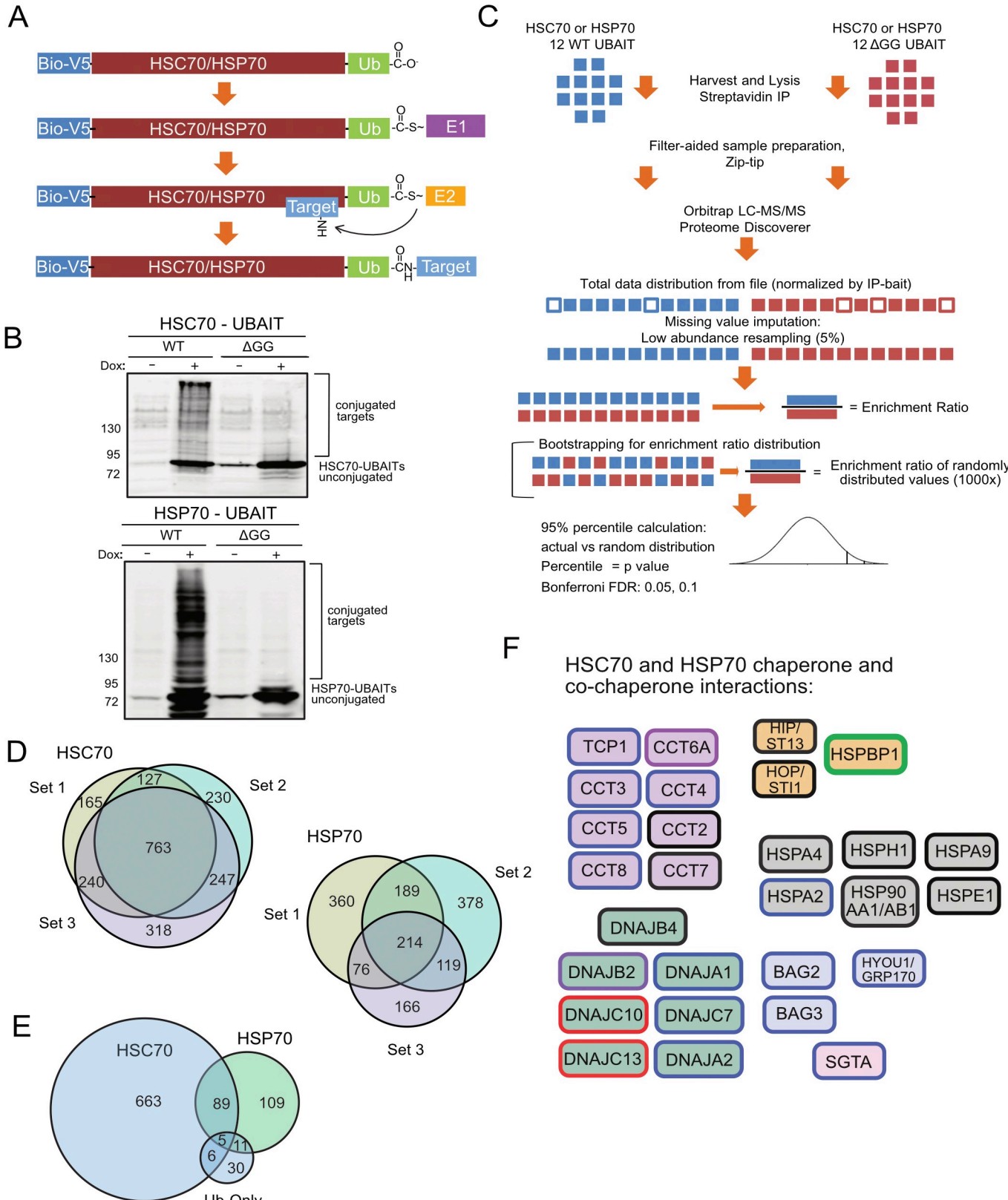

**Fig 1. Global identification of HSPA1A and HSPA8 clients using UBAIT in human cells.** (A) Schematic diagram of biotin/V5-HSP70/HSC70 UBAIT conjugation to bound targets. (B) Western blot of HSP70 and HSC70 UBAITs after streptavidin isolation from human U2OS cells treated with doxycycline

(Dox) (1 ug/mL) for 3 days or untreated. (C) Schematic of experimental and statistical workflow used for the analysis of UBAIT-conjugated targets. Twelve wild-type or ΔGG samples are first analyzed by LC-MS/MS using label-free quantification, then missing values are imputed by low abundance resampling (5%). Wild-type and ΔGG values are used to calculate an enrichment ratio and also are bootstrapped to generate random enrichment ratio distribution (1,000×). The actual enrichment ratio is compared with the bootstrapped distribution to obtain *p*-values, which are corrected for multiple testing hypotheses using the Benjamini-Hochberg method. (D) Venn diagrams of HSC70 and HSP70 targets identified from 3 independent experiments, each with *N* = 12, all with K48R ubiquitin fusions. (E) Venn diagram of proteins identified as significantly enriched from all three experiments compared with the ubiquitin-only control. (F) Co-chaperones and known chaperone interactors identified from (D). All factors shown were identified at FDR 0.05. Black outline: enriched for HSC70 and HSP70 binding in all 3 sets for both chaperones; red outline: enriched for HSC70 in all 3 sets, no enrichment for HSP70; blue outline: enriched for HSC70 in all 3 sets, enriched for HSP70 in at least one set; green outline: enriched for HSP70 in all 3 sets, no enrichment for HSC70; purple outline: enriched for HSP70 in all 3 sets, enriched for HSC70 in at least one set. Fill colors show chaperones (gray), DNAJ proteins (green), nucleotide exchange factors (blue), other co-chaperones (orange or pink), and chaperonins (purple). Bio-V5, biotin-V5 tag; DNAJ, heat shock protein 40; FDR, false discovery rate; HSC, heat shock cognate; HSP, heat shock protein; IP, immunoprecipitation; LC-MS/MS, liquid chromatography–mass spectrometry; Ub, ubiquitin; UBAIT, ubiquitin-activated interaction trap; WT, wild-type.

identified proteins in wild-type and control ΔGG isolations to generate an enrichment ratio and used a bootstrapping method to calculate *p*-values for likelihood of interaction (see Materials and methods). Lastly, we identified the most significant interactions using modified Benjamini-Hochberg correction to control the false positive rate [27]. The 24 isolations were analyzed together in one group, and two additional biological repeats of this procedure were performed, each with 12 wild-type and 12 ΔGG isolations. All the samples were generated from a single U2OS cell line over the course of approximately 2 weeks, and the samples for each group were collected and analyzed independently. Using this workflow, we identified an average of 1,409 interactors for the three HSC70 UBAIT experiments, with 763 proteins common to all three sets (Fig 1D, 1E, and S1 Data). The HSP70 UBAITs yielded fewer binding partners, with an average of 772 potential interactors, with 214 proteins shared among all three sets. Examples of HSC70 and HSP70 target enrichment data shown in S3 Fig. We also note here that HSP70 UBAIT expression, as regulated by an inducible promoter in this system, is higher than the endogenous protein that is normally found at high levels only in response to heat or other forms of proteotoxic stress [6].

The UBAIT strategy relies upon cellular enzymes to conjugate the fused ubiquitin to targets [25]. To ensure that the binding partners we identified are actually specific to the chaperones and not nonspecifically associating with ubiquitin, we created biotin-V5–tagged ubiquitin-only constructs (wild-type and ΔGG) to serve as an additional control for the experiments and performed one set (12 wild-type and 12 ΔGG samples) of isolations, which identified 52 unique proteins. Comparison of this set with the chaperone UBAIT data showed a relatively small number of the binding partners of HSC70 and HSP70 were shared with biotin-V5–ubiquitin only (Fig 1E, S1 Data), 16 from the HSC70 list and 11 from HSP70, with 5 shared between them. Although many proteins can be ubiquitinated in cells [28], the strict statistical thresholds used here for interactors likely prevents the large number of ubiquitinated targets from being recovered at high levels. It is clear from this comparative analysis that the vast majority of the interacting proteins identified with HSC70 and HSP70 are actually chaperone-binding factors.

Analysis of the proteins identified as interactors of HSC70 and HSP70 shows many known co-chaperones and binding partners of these chaperones (Fig 1F). Those that are common to both enzymes include the HSP90AA1/AB1 chaperones that fold substrates cooperatively with HSP70 family enzymes [29] as well as the "bridging" cofactor HSP70-HSP90 organizing protein (HOP/STI1) that binds to HSP70 and HSP90 [30]. We also recovered HSC70-interacting protein (HIP/ST13) that regulates chaperone activity [31] and several nucleotide exchange factors for the HSP70 family, including Bcl-2–associated athanogene domain proteins (BAG2 and BAG3), as well as several members of the DnaJ homolog subfamily of HSP40 proteins that

regulate the catalytic cycle of HSP70 enzymes: DNAJA1, DNAJC7, DNAJC10, DNAJC13, DNAJB2, and DNAJB4. We also identified interacting HSP70 family members that regulate substrate binding and release: HSPH1, HSPA4, and GRP170 (HYOU1) [32], and other compartment-specific members of the HSP70 family (HSPA2, HSPA9B, and HSPE1) and several subunits of chaperonin-containing TCP-1 (CCT proteins) [33,34].

Although several cofactors were observed to bind to both HSC70 and HSP70, some of the interacting cofactors were specific for HSC70. For instance, the DNAJC10 and DNAJC13 HSP40 proteins, which promote ATP hydrolysis by HSP70 chaperones, were only identified with statistically significant binding for HSC70, suggesting that the occupancy of these cofactors with the constitutive chaperone is much greater than with the heat-inducible form. In contrast, the nucleotide exchange factor HSPBP1 is only found with HSP70, not with HSC70, suggesting that this regulatory protein binds stably to the heat-inducible form.

Besides known co-chaperones and adaptors for HSP70 enzymes, we observed a large number of cellular proteins that do not have known association with HSC70 or HSP70 and are likely to be client proteins (S1 Data). The interactors are predicted to be localized in many cellular compartments, consistent with the role of HSC70 and HSP70 in promoting the folding and localization of cytosolic, nuclear, and membrane proteins with diverse functions [6]. For instance, the most enriched gene ontology–based biological functions of the HSC70 and HSP70 UBAIT target sets include RNA-binding factors, metabolic enzymes, structural proteins, and chromosome-associated factors (S4 Fig).

The high degree of overlap between the 3 UBAIT experiments performed, particularly for HSC70, indicates that the UBAIT strategy combined with the sample isolation, mass spectrometry, and statistical analysis yields reproducible and statistically significant chaperone-binding targets. These results can be considered as an HSC70/HSP70 chaperone-binding snapshot of this cell line at a specific point in time; however, we do not suggest that these define an immutable client repertoire. We have observed that client-binding patterns change over time within the same cell line, suggesting that chaperone clients are sensitive to environmental as well as intracellular stress conditions, as discussed in more detail below.

The HSP70 family of proteins are part of a larger system of protein quality control and have been shown to regulate both proteasome-mediated degradation of proteins as well as chaperone-mediated autophagy (CMA) [8]. The experiments in Fig 1 were performed with a K48R version of ubiquitin in the UBAIT experiments to block degradation of the fusion protein [35], but we were also interested in how this mutation affects the recovery of HSC70 and HSP70 clients. We performed the UBAIT experiment with K48 wild-type versions of ubiquitin and found that the number of binding partners identified in the HSP70 UBAIT did not change significantly from the three independent experiments that were performed with K48R ubiquitin (S5 Fig). However, with HSC70, we were unable to identify targets with statistically significant binding (only 1 protein identified after filtering for nonspecific interactions and controlling the false discovery rate [FDR]). These results may suggest that client proteins associated with HSC70 and ubiquitin constitute a potent degradation signal, while the equivalent complex with HSP70 does not. These results are consistent with reports showing that HSC70 is required for efficient proteasome-mediated degradation of some clients [36,37], and evidence for HSC70/HSP70 proteins in chaperone-assisted ubiquitin–proteasome degradation [38].

## The V438F HSC70 substrate-binding mutation separates clients from other binding partners

HSP70 enzymes associate with many binding partners: co-chaperones, clients, and also other complexes that are not in these categories [39]. To assess what subset of binding partners are

likely to be clients, we performed a UBAIT experiment with constitutive HSC70, comparing wild-type to the substrate binding mutant V438F. This residue is located in the hydrophobic substrate-binding pocket, and the V438F mutation has been shown to substantially reduce substrate binding [40]. We performed 6 UBAIT isolations each from wild-type and V438F HSC70 (K48R) UBAIT-expressing cells, with an additional 6 isolations from wild-type and V438F ΔGG controls. We recovered fewer targets in this experiment compared with the larger analysis shown in Fig 1 but still obtained 239 significant targets after FDR correction for the wild-type HSC70 UBAIT and 251 for the V438F UBAIT (S6 Fig and S2 Data). A total of 128 targets were shared between these groups; these included the major co-chaperones: HSP90, HSPA1A/B, STIP1, ST13, and HSPA4, as well as DNAJB2 and small glutamine rich tetratrico-peptide repeat containing alpha protein (SGTA). A total of 111 binding partners of the wild-type enzyme were not recovered with the V438F version of the UBAIT. This group does include some co-chaperones, including the J domain proteins DNAJC7 and DNAJC13. Unlike canonical DnaJ family co-chaperones, which bind to Hsp70s through their J domains and the nucleotide-binding domain of HSP70 [41], DNAJC7(TPR2) associates with HSP70 through interactions between the TPR domains in DNAJC7 and the substrate-binding domain of HSP70 [42], which may explain why the binding of this J domain is sensitive to the V438F mutation. DNAJC13 (RME8) is also in this group of V438F-sensitive binding targets, perhaps due to its reported affinity for the ADP-bound (substrate-associated) form of HSC70 [43], in contrast to the ATP-bound HSC70 binding preference of most J proteins [44,45]. Besides these co-chaperones, the 111 V438F-sensitive binding partners likely include many client proteins. This group does not show any significant differences from the total proteome with respect to charge, but an analysis of the sequences using the TANGO and WALTZ algorithms [46,47], which predict propensity for amyloid and beta-sheet aggregation based on experimentally derived parameters, shows a significant difference between the V438F-sensitive targets of HSC70 in comparison to the targets bound by the V438F form of the UBAIT (S6 Fig). If the targets shared between these groups are removed, these differences are even more extreme, with the proposed client group showing TANGO, WALTZ, and length averages that are 40% to 70% higher than the group of targets that is bound only to the mutant UBAIT. Thus, one of the clear differences between the proposed client and non-client targets is propensity for aggregation.

The V438F mutant HSC70 UBAIT also preferentially bound to 123 targets that were not recovered at significant levels with the wild-type UBAIT (S6 Fig). Examination of this list shows that several components of the chaperonin T-complex protein (TCP)1-ring complex (TRiC) complex are represented here (TCP1, CCT7, and CCT4). This complex is known to bind to HSC70 [48,49] and was shown to associate through the nucleotide-binding domain of the chaperone [50]; thus, it is perhaps not surprising that the association is still observed with the mutant UBAIT. Why this is observed with the mutant and not the wild-type UBAIT is not clear, although it is possible that because a limited pool of UBAIT exists, the elimination of client binding through the substrate binding cleft allows for a larger subset of the UBAIT to associate with non-clients. Further analysis of this V438F-specific binding fraction shows that several proteins are known substrates for CMA, a process by which HSC70 recognizes substrates through a conserved motif (KFERQ) and promotes its relocalization to the surface of lysosomes, where it is internalized and degraded [51]. The KFERQ binding motif does not bind to the HSC70 substrate binding cleft in the same way as canonical substrates [52] and thus is likely not to be as affected by the V438F mutation. Glyceraldehyde 3-phosphate dehydrogenase (GAPDH), perilipin 3 (PLIN3), dihydropyrimidinase like 2 (DPYSL2), and annexin A1 (ANXA1) are all known to be regulated through CMA [53–56], so these and potentially other factors on this list of V438F-specific binding partners may be CMA targets. Other factors

in this set are known to be stable binding partners of HSC70, likely through other interfaces [57–68]. Overall, this analysis of the V438F interactome suggests that about half of the UBAIT targets are bound in a V438-sensitive manner, while half are not, with the latter group including co-chaperones as well as binding partners in other complexes.

## Comparison with BioID

To validate the results from the UBAIT-based interaction screen, we also analyzed HSP70 clients using an orthogonal method—BioID [69,70]. This strategy involves fusion of a bait protein to the biotin ligase BirA, which results in biotinylation of proteins bound to the bait, although in this case, the fusion is with the N terminus of the bait protein. Transient interactions are captured with this method, similar to the UBAIT fusions, and targets are also recovered with high stringency biotin-streptavidin interactions. Here, we inducibly expressed an HSP70 fusion with *Aquifex aeolicus* biotin ligase (BioID2) [70], as well as biotin ligase alone, and performed 12 isolations for each with streptavidin beads using the same isolation procedures as were used in the UBAIT experiments. After controlling the FDR with a modified Benjamini-Hochberg method and excluding the targets isolated with biotin ligase alone, we identified 438 polypeptides as binding partners of HSP70 (S3 Data). Examples of targets recovered with the BioID2-HSP70 fusion are also shown in S7 Fig. Of these 438 targets, 252 (58%) were also found with the HSP70 UBAIT fusion in at least one of the 3 sets of UBAIT experiments performed, indicating that there is reasonable overlap between the clients identified by these methods. Co-chaperones and chaperone-associated factors were also well-represented in this group, including HSPH1, HSPA2, HSPA4, HSPA5/BIP, HSPA9, HSPE1, STIP1/Hop, ST13, chaperonins, DNAJ proteins, BAG2, and BAG3 (S7 Fig).

## HSP70 and HSC70 preferentially bind to newly synthesized proteins

HSC70 and HSP70 orthologs are known to be associated with translating ribosomes [11,19,71]. To determine if the clients of HSP70 and HSC70 are biased toward newly translated proteins, we used a stable isotope labeling with amino acids in cell culture (SILAC)-based approach to distinguish between newly made proteins and older proteins in the cells. To test this strategy, U2OS cells were grown for at least 5 doublings in media containing standard L-lysine and L-arginine, and then switched into heavy ($^{13}C$, $^{15}N$) L-lysine and L-arginine media for 1, 2, or 3 days before harvesting (Fig 2A, top). Doxycycline induction of chaperones was started at time "0" to ensure that all chaperone UBAIT protein was expressed in heavy isotope. Total proteins in the lysates were analyzed for light/heavy ratios in tryptic peptides, which showed that the overall light/heavy ratios decreased with the length of heavy media pulse, as expected, with some proteins showing less change in the ratio over time (long-lived), whereas other proteins changed at a higher than average rate (short-lived) (Fig 2A).

To use this system in the context of our UBAIT chaperone isolations, U2OS cells expressing tagged HSC70 or HSP70 UBAITs were incubated with light media for at least 5 doublings before induction of UBAIT expression with doxycycline (Fig 2B, top panel). The cells were changed to heavy media at the time of induction and grown for either 1 or 2 days before harvesting. Chaperone UBAIT isolation was then performed as in Fig 1, with 12 wild-type UBAIT isolations performed for each chaperone. Binding partners were analyzed for light/heavy ratios by mass spectrometry, and the total patterns of these ratios for HSC70 and HSP70 validated partners show an average of 0.95 and 0.63 light/heavy log ratios for a 1-day pulse, and −0.31 and −1.22 for a 2-day pulse, respectively ("actual" distributions in Fig 2B). Because every protein has a unique turnover rate, we also calculated the expected light/heavy ratio for each protein for the duration of the pulse, based on the results from the total lysates in our 1- and 2-day

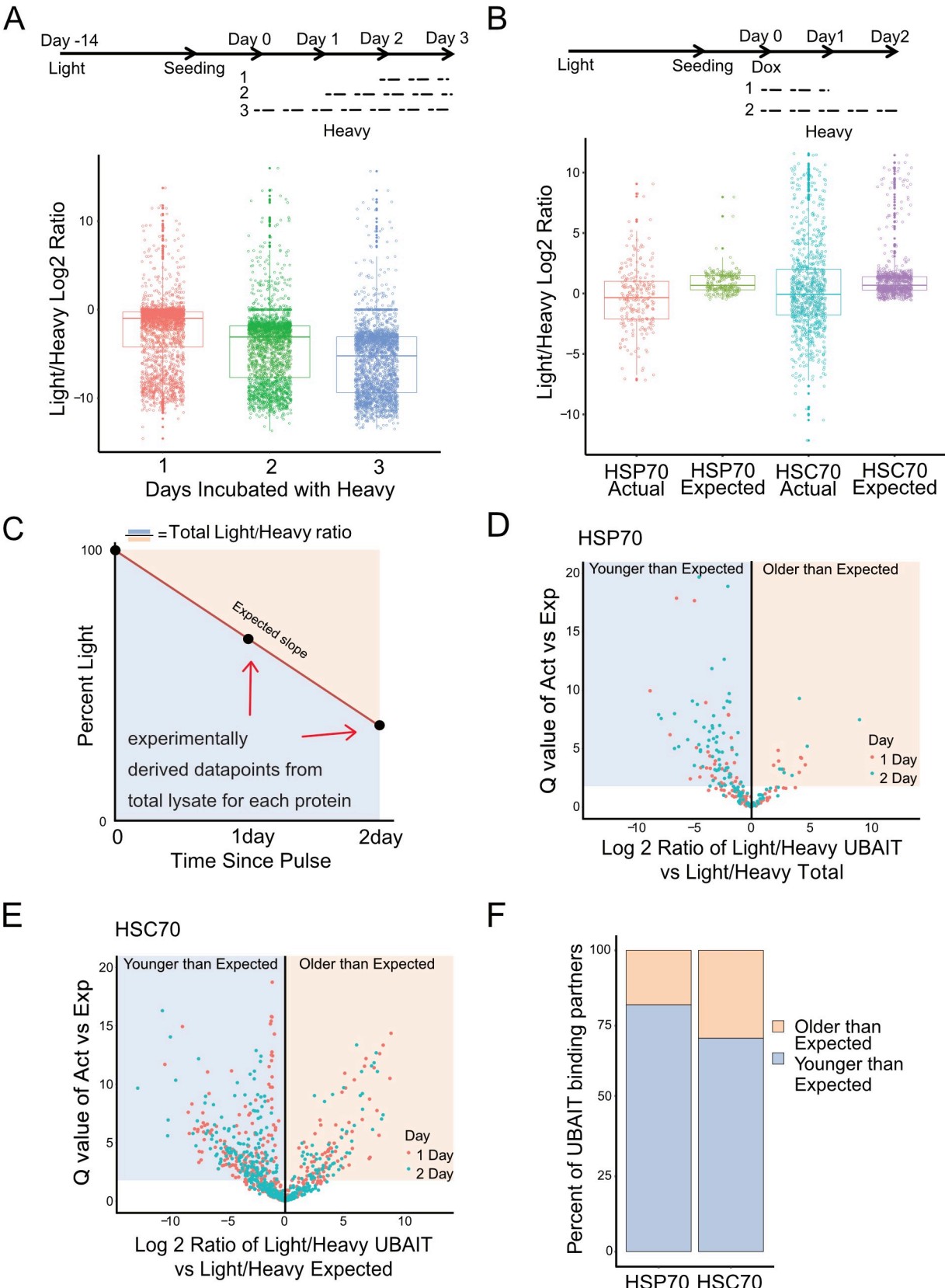

**Fig 2. HSC70 and HSP70 binding partners are enriched for newly synthesized proteins.** (A) Top: schematic diagram of SILAC pilot experiment performed with U2OS cells; cells were grown in light arginine and lysine media prior to experiment for least 14 days. Media was changed to heavy arginine and lysine for 1, 2, or 3 days before harvesting. Protein in total lysates was analyzed for light/heavy ratios by label-free mass spectrometry and polypeptides identified from all 3 samples are shown. (B) Top: Schematic view of SILAC UBAIT experiment used for 12 wild-type UBAIT samples from HSC70 and HSP70 at each time point. Bottom: comparison of light/heavy ratios of HSP70 and HSC70 targets from the SILAC UBAIT isolations, only showing targets verified from UBAIT experiments performed in Fig 1. (C) Schematic showing method for calculation of expected light/heavy ratios. Total cell lysate light/heavy ratio data were used to experimentally derive two data points for each enriched protein identified. These two points were used to calculate a slope to obtain a theoretical expected light/heavy ratio for each UBAIT target. (D,E) Volcano plot of proteins enriched for HSP70/HSC70 association by comparing the actual ratio to the expected ratio for each protein, with significance (Q value = −log10 of *p*-value) on the *y* axis. *p*-values were calculated using a one-sample *t* test, comparing the 12 "actual" ratios to the "expected" ratio. Proteins to the left of the *y* axis are proteins that have lower light/heavy ratios than expected, while proteins to the right are proteins that have higher light/heavy ratios than expected. (F) Summary of results from (D) and (E). Protein counts from the lower-than-expected category or the higher-than-expected category are summarized. See also S6 Data. Act, actual; Exp, experimental; HSC, heat shock cognate; HSP, heat shock protein; SILAC, stable isotope labeling with amino acids in cell culture; UBAIT, ubiquitin-activated interaction trap.

SILAC experimental dataset, with an assumption that the heavy isotope incorporation rate is linear over 2 days (Fig 2B and 2C). This "expected" ratio for each protein comes from the total lysate data and therefore represents the overall turnover rate for that protein in the cell, whereas the UBAIT target data show the ratio specifically for the polypeptides captured by the UBAIT tagged chaperone.

Comparison of the UBAIT actual light/heavy ratios versus the expected light/heavy ratios for HSC70 and HSP70 UBAIT binding partners shows that there are more UBAIT partners that are significantly younger (heavy) relative to total protein present in the lysate (Fig 2D, 2E, 2F and S4 Data). Shaded areas in Fig 2D and 2E show proteins with significantly different ratios in the actual versus expected values, the majority of which are younger than expected. This preference is observed for both HSC70 and HSP70 and is consistent with the idea that a majority of unfolded or misfolded targets are bound by these chaperones either at the ribosome or shortly after translation. Both short-lived and long-lived proteins were observed in the UBAIT datasets, and thus there was no obvious preference for polypeptides showing rapid turnover.

## HSC70 and HSP70 association with proteins lacking binding partners

One important biological scenario that requires HSC70/HSP70 chaperone function is the maintenance of proper stoichiometry within multicomponent complexes. "Orphan" proteins that lack essential binding partners are rapidly degraded in eukaryotes in a manner that requires HSP70 family enzymes [72–75], and the maintenance of stoichiometry within multi-component complexes in eukaryotes was recently suggested to occur through a co-translational process [72,76]. To test whether HSC70 or HSP70 specifically interacts with proteins in multi-subunit complexes that are destined to be degraded in the absence of their obligate binding partners, we chose a multi-subunit complex in which one member of the complex is dependent on at least one other member for its stability. Meiotic recombination 11 protein (Mre11), a DNA repair factor, is absolutely essential for the stability of its binding partner, Rad50 [77]. Rad50 is not observed consistently as an HSC70 or HSP70 UBAIT binding partner under normal growth conditions (S1 Data). We reduced levels of Mre11 by small hairpin RNA (shRNA) depletion in U2OS cells and found that this reduced not only Mre11 but also the total amount of Rad50 protein, as expected (Fig 3A). We performed 6 UBAIT isolations of tagged HSC70 and HSP70 from cells with the Mre11 shRNA expressed, and found that levels of Rad50 bound to HSC70 increased 3.2-fold compared with samples without Mre11 depletion. This was not the case for HSP70, which showed no increase in Rad50 association. These results collectively suggest that, in this case, HSC70 specifically binds to an orphan protein that is missing its obligate binding partners, while the heat-inducible HSP70 does not play this role.

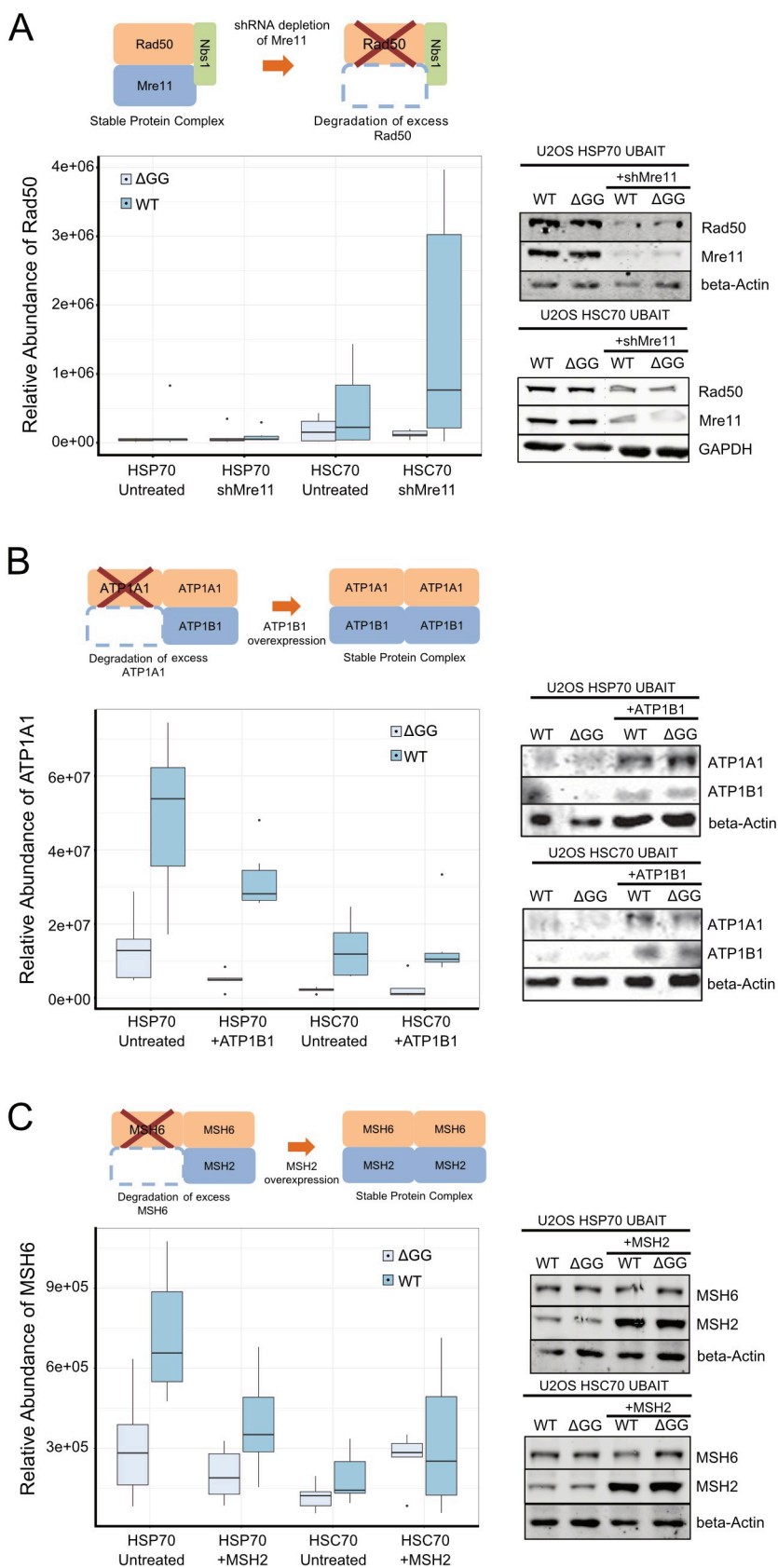

**Fig 3. HSC70 and HSP70 association with proteins lacking binding partners.** (A) Top: schematic diagram of the Mre11/Rad50/Nbs1 complex showing that loss of Mre11 induces loss of Rad50. HSP70 and HSC70 UBAIT isolations were performed with or without depletion of endogenous Mre11 by shRNA ($N = 6$). Levels of Rad50 associated with HSP70 or HSC70 UBAITs were measured using label-free quantification. Western blot showing levels of Mre11, Rad50, and control proteins GAPDH and beta-actin in UBAIT cells with Mre11 shRNA, as indicated. (B) Top: schematic diagram of the ATP1A1/ATP1B1 complex showing that insufficient levels of ATP1B1 lead to loss of ATP1A1, while overexpression of ATP1B1 restores the complex and ATP1A1 stability. HSP70 and HSC70 UBAIT isolations were performed with or without overexpression of ATP1B1 ($N = 6$). Levels of ATP1A1 associated with HSP70 or HSC70 UBAITs were measured using label-free quantification. Western blot showing levels of ATP1A1, ATP1B1, and beta-actin in UBAIT cells with ATP1B1 expression, as indicated. (C) Top: schematic diagram of the MSH2/MSH6 complex showing that insufficient levels of MSH2 lead to loss of MSH2, while overexpression of MSH2 restores the complex and MSH6 stability. HSP70 and HSC70 UBAIT isolations were performed with or without overexpression of MSH2 ($N = 6$). Levels of MSH6 associated with HSP70 or HSC70 UBAITs were measured using label-free quantification. Western blot showing levels of MSH2, MSH6, and beta-actin in UBAIT cells with MSH2 expression, as indicated. See also S6 Data. ATP1A1, ATPase, Na+/K+ transporting subunit alpha 1; ATP1B1, ATPase, Na+/K+ transporting subunit beta 1; GAPDH, glyceraldehyde 3-phosphate dehydrogenase; HSC, heat shock cognate; HSP, heat shock protein; Mre11, Meiotic recombination 11 protein; MSH, MutS homolog; Nbs1, Nijmegen Breakage Syndrome protein 1/Nibrin; shRNA, small hairpin RNA; UBAIT, ubiquitin-activated interaction trap; WT, wild-type.

We also were interested in testing whether expression of known obligate partners could reduce levels of HSP70/HSC70 client binding. For this, we chose a protein that was consistently identified as one of the highest-ranked binding partners of HSP70 and HSC70, the Na+/K+ transporting subunit alpha 1 (ATP1A1) subunit in the cation transport ATPase family. The stability of ATP1A1 is dependent on its binding partner, Na+/K+ transporting subunit beta 1 (ATP1B1), in the heteromeric ATPase complex [78]. ATP1B1, in contrast to ATP1A1, is not enriched for either HSC70 or HSP70 in our UBAIT experiments (S1 Data). We hypothesized that the level of ATP1A1 expressed in this cancer cell line exceeds that of its obligate partner ATP1B1, and thus ATP1A1 is recognized by HSP70/HSC70 constitutively in these cells. In this case, we should be able to reduce the level of ATP1A1 binding to the chaperones by overexpression of ATP1B1. To test this, we overexpressed ATP1B1 by transient expression in the HSC70 and HSP70 UBAIT cells and monitored the levels of both proteins. As expected, overexpression of ATP1B1 increased the stability of ATP1A1 (Fig 3B). In addition, the levels of ATP1A1 observed in the 6-sample UBAIT isolation of HSP70 decreased 3.6-fold with ATP1B1 expression compared with the untreated group of samples. Unlike the example with Mre11 and Rad50, in this case the levels of ATP1A1 bound to HSC70 did not change.

To further characterize the relationship of orphan proteins and chaperone binding, we chose the MutS homolog 2–6 (MSH2-MSH6) heterodimeric complex, a mismatch repair recognition enzyme in which MSH6 is strongly dependent on MSH2 for stability [79]. Like ATP1A1, MSH6 is recovered as an interactor with UBAIT-tagged chaperones, but MSH2 is not (S1 Data). We overexpressed MSH2 in HSC70 and HSP70 UBAIT cells, performed a 6-replicate experiment for each tagged chaperone, and observed a 1.9-fold reduction in MSH6 levels associated with the HSP70 UBAIT specifically, but not with HSC70 (Fig 3C), similar to the result with ATP1A1 and ATP1B1. Taken together, these depletion and overexpression experiments suggest that HSC70 and HSP70 are monitoring levels of orphan proteins and that, in some cases, there seem to be specific interactions with one but not both of the chaperones. The experiments in which we have partially restored missing binding partners show that these interactions can be modulated in predictable ways by changing the expression of their obligate partners.

## Expression of a misfolded protein changes the landscape of HSC70 binding partners

Efficient control of protein homeostasis is critical for all cells but particularly for some tissues, for instance neurons and other nondividing cells in the mammalian brain [80]. The

consequences of protein misfolding in neuronal cells can include neurodegeneration, in which case misfolded proteins are implicated in each disorder that have pathological effects in specific neuronal populations [81]. We tested the binding of HSC70 and HSP70 to one of these factors, the superoxide dismutase mutant A4V (SOD1 A4V) associated with a subset of familial ALS and the cause of highly penetrant and rapid motor neuron loss [82]. The A4V mutation in SOD1 promotes the formation of misfolded but soluble oligomers that are thought to be associated with neurotoxicity [83]. Here, we expressed SOD1(A4V) co-translationally with green fluorescent protein (GFP) (GFP-P2A-SOD1) using a BacMam virus in our HSC70 and HSP70 UBAIT cells at levels approximately 10-fold higher than endogenous SOD1 and isolated targets from 12 samples for each chaperone (with 12 ΔGG controls). We compared these results to an equal number of samples in which GFP expression alone was induced. We observed significantly increased binding of SOD1 to both chaperones with SOD1(A4V) over-expression under these conditions (Fig 4A).

We expected that the expression of a misfolded protein might cause one or both chaperones to exhibit lower levels of binding to endogenous clients because of the acquired association with SOD1(A4V). We did see this, with both HSC70 and HSP70 showing a loss of 44% of the bound partners observed in the absence of SOD1(A4V) expression. Unexpectedly, we also observed new partners upon SOD1(A4V) overexpression, approximately half as many as were lost (Fig 4B and S5 Data). We examined this new set of acquired binding partners (206 for HSP70 and 227 for HSC70) and found that the newly bound partners of HSC70 were significantly increased for both polypeptide length and for predicted disorder based on the TANGO algorithm, which estimates intermolecular beta-sheet protein aggregation [46](Fig 4C). The binding partners of HSC70 and HSP70 in the absence of misfolded protein expression do not exhibit any significant differences in these parameters in any of the other trials compared with the total lysate.

We also utilized a method to isolate detergent-resistant protein aggregates [84] from the cells expressing GFP only or GFP plus SOD1(A4V) to determine if there are any changes in protein aggregation. We performed isolations and mass spectrometry–based quantitation from 3 biological replicates with cells expressing GFP only or GFP plus SOD1(A4V) and identified a total of 2,433 proteins in the aggregate fractions (Fig 4D). The majority of these (1,520) did not change significantly with SOD1(A4V) expression, but we found that 620 proteins were recovered at reduced levels, while 293 were more enriched with expression of the misfolded SOD1. These 293 proteins were not related to the proteins identified in our UBAIT isolations but, similar to the HSPA8 UBAIT targets, they did show a bias toward proteins with higher predicted disorder (TANGO score) [46].

To determine if the apparent change in binding partners observed in the UBAIT experiment was due to the high level of SOD1(A4V) expression, we also repeated this experiment with lower levels of SOD1(A4V) expression by reducing the amount of BacMam virus, again analyzing 12 wild-type and 12 ΔGG controls (Fig 4E). Here, SOD1(A4V) expression levels were only 1.6-fold over the endogenous protein level. SOD1 was still observed binding to both HSC70 and HSP70, albeit with lower overall efficiency compared with the experiment with higher SOD1(AV4) expression. Nevertheless, we still observed 36% loss of clients with HSC70 and 4% loss with HSP70 under these conditions (Fig 4F). In addition, we still observed a gain in novel HSP70 and HSC70 clients, and the HSC70 clients again specifically exhibited a higher predicted level of intrinsic disorder (Fig 4G). Here, the proteins bound by HSC70 in response to SOD1(A4V) expression exhibited not only higher TANGO scores for beta-sheet aggregation propensity but also higher WALTZ scores, which estimate amyloid-forming potential in proteins based on experimentally determined physical properties [47].

Collectively, the results with SOD1(A4V) expression show that expression of a misfolded protein, even at low levels, can cause dramatic shifts in chaperone binding patterns for both

A

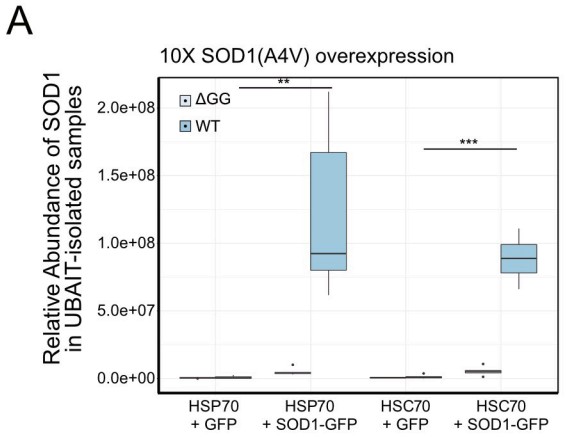

B

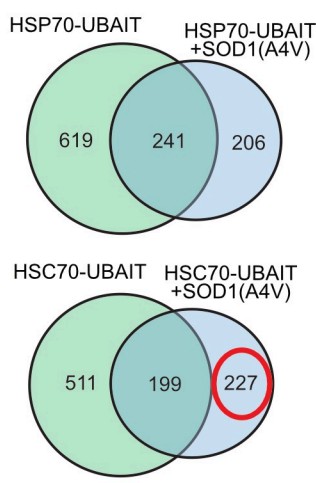

HSP70-UBAIT | HSP70-UBAIT +SOD1(A4V)
619 | 241 | 206

HSC70-UBAIT | HSC70-UBAIT +SOD1(A4V)
511 | 199 | 227

C

|  | HSP70 | | |
| --- | --- | --- | --- |
|  | Untreated | SOD1 A4V OE | P-Value |
| WALTZ | 1210.051 | 1138.808 | 0.2635 |
| TANGO | 2311.042 | 2318.57 | 0.4884 |
| AA Length | 624.12 | 617.82 | 0.4501 |
|  | HSC70 | | |
|  | Untreated | SOD1 A4V OE | P-Value |
| WALTZ | 1122.793 | 1278.066 | 0.1156 |
| TANGO | 2093.333 | 2548.502 | 0.04226 |
| AA Length | 560.28 | 708.43 | 0.01212 |

D

detergent-resistant aggregates:

620 proteins showing reduced aggregation with SOD1(A4V) expression (p<0.05 only)

293 proteins showing increased aggregation with SOD1(A4V) expression (p<0.05 only)

1520 unchanged with SOD1(A4V) expression (p>0.05)

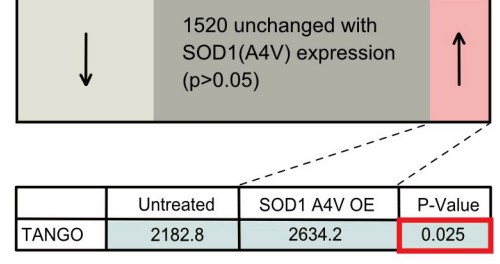

|  | Untreated | SOD1 A4V OE | P-Value |
| --- | --- | --- | --- |
| TANGO | 2182.8 | 2634.2 | 0.025 |

E

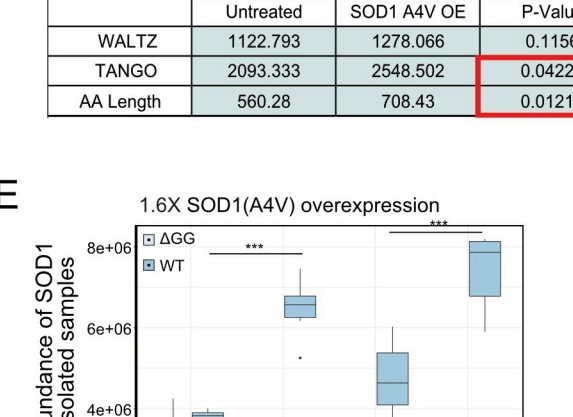

G

|  | HSP70 | | |
| --- | --- | --- | --- |
|  | Untreated | SOD1 A4V OE | P-Value |
| Waltz | 1117.295 | 1199.392 | 0.1818 |
| Tango | 2285.333 | 2509.054 | 0.146 |
| AA Length | 609.9072 | 632.7357 | 0.2982 |
|  | HSC70 | | |
|  | Untreated | SOD1 A4V OE | P-Value |
| Waltz | 1133.681 | 1333.127 | 0.04776 |
| Tango | 2134.442 | 2604.228 | 0.03272 |
| AA Length | 604.804 | 642.7912 | 0.2229 |

F

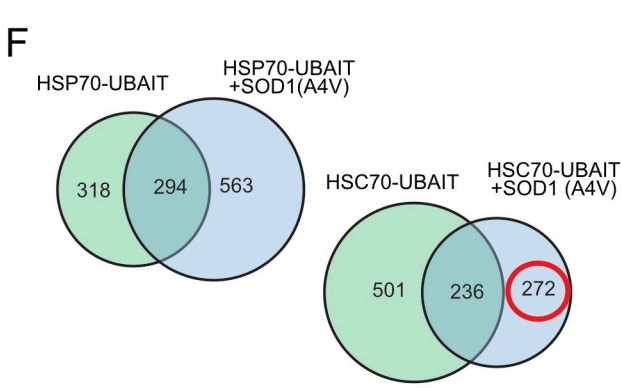

HSP70-UBAIT | HSP70-UBAIT +SOD1(A4V)
318 | 294 | 563

HSC70-UBAIT | HSC70-UBAIT +SOD1 (A4V)
501 | 236 | 272

**Fig 4. Expression of a misfolded protein changes the landscape of HSC70 binding partners.** (A) HSP70 and HSC70 UBAIT isolations were performed in U2OS cells with either GFP or SOD1(A4V)-P2A-GFP high-level overexpression (N = 6); levels of SOD1 associated with HSP70 or HSC70 UBAITs were measured using label-free quantification. (B) Venn diagrams showing proteins identified as enriched with either HSP70 or HSC70 UBAITs; red circle indicates targets gained with SOD1(A4V) expression in HSC70 UBAIT cells. (C) Summary of average WALTZ [47] and TANGO [46] scores as well as polypeptide length of proteins enriched with UBAITs in cells expressing GFP or SOD1(A4V)-P2A-GFP. Welch's *t* test was used to compute *p*-values, and red box indicates significant *p*-values corresponding to targets gained with SOD1(A4V) expression in HSC70 UBAIT cells. (D) Detergent-resistant protein aggregates were isolated and analyzed by mass spectrometry from UBAIT-expressing cells with either GFP or SOD1(A4V) expression (N = 3). Summary of proteins identified that increased, decreased, or were statistically indistinguishable in aggregate fraction with SOD1(A4V) expression. Welch's *t* test was used to compute *p*-values. (E) HSP70 and HSC70 UBAIT isolations were performed as in (A) but with lower-level overexpression (approximately 2-fold compared with endogenous)(N = 6). Levels of SOD1 associated with HSP70 or HSC70 UBAITs were measured using label-free quantification. (F) Venn diagrams showing proteins identified as enriched with either HSP70 or HSC70 UBAITs. (F) Summary of average WALTZ [47] and TANGO [46] scores as well as polypeptide length of proteins enriched with UBAITs in cells expressing GFP or SOD1(A4V)-P2A-GFP. Welch's one-tailed *t* test was used to compute *p*-values, and red box indicates significant *p*-values corresponding to targets gained with SOD1(A4V) expression in HSC70 UBAIT cells. See also S6 Data. AA, amino acid; GFP, green fluorescent protein; HSC, heat shock congnate; HSP, heat shock protein; OE, overexpressed; SOD1, superoxide dismutase 1; UBAIT, ubiquitin-activated interaction trap; WT, wild-type.

HSC70 as well as HSP70. These include not only association with the misfolded species but also a reorganization of client binding. For HSC70 specifically, there appears to be a shift to proteins with different physical features that include factors with intrinsically disordered regions and longer polypeptides.

## Discussion

### Employing UBAIT to identify a complex set of interactors

We have established in this work that the UBAIT fusion system can be used to profile complex interactomes. This method has previously been used to identify partners of bait proteins that have relatively few targets [85,86], but here we show that this strategy can be used on a large scale to identify hundreds of interacting partners. UBAITs have the advantage that even transient interactions can be captured in covalent association between the bait and the binding protein. This is useful in normalization between samples, as the wild-type and ΔGG forms of the fusion protein are both isolated and quantitated. In contrast, it is more difficult to normalize samples with BioID fusions, as the BioID fusion protein is not reliably isolated during the procedure. The UBAIT ΔGG isolations also provide an excellent negative control in these experiments because they express the identical bait protein, only lacking the last two glycine residues of ubiquitin. This provides a stringent comparison for nonspecific interactions and allows for confident identification of even low-level interactors.

Mass spectrometry reliably identifies the most abundant polypeptides in a biological sample, but low-level targets may be missed, depending on the mass/charge ratio of individual peptides and the mixture of other proteins present during the analysis [87]. For these reasons, failure to identify a given peptide or protein in a complex mixture is not necessarily a confident indicator of its absence. In this work, we used a large number of replicate isolations (12 per set), with an equal number for the ΔGG controls, and also performed 3 sets per chaperone in order to acquire sufficient data for low-level target identification. We used a bootstrapping method to determine the statistical probability for likelihood of each outcome, only taking those exceeding the 95% confidence interval. This strategy, combined with modified Benjamini-Hochberg methods to control the FDR, allows us to identify low-level interactors (see Materials and methods for details).

### HSP70 and HSC70 chaperones as nonredundant monitors of stress states

HSP70 and HSC70 targets have historically been very challenging to identify and quantify because of the large number of potential client interactions, as well as the transient nature of

these interactions. We have demonstrated in this study that it is possible to globally interrogate HSP70/HSC70 chaperone binding partners in human cells and that these interactions change in response to physiologically relevant sources of stress. We find that both HSP70 and HSC70 clients are biased toward newly synthesized proteins, but they are largely nonoverlapping in their binding preferences; thus, these enzymes should not be considered identical in function or associations. The changes in both HSC70 binding partners as well as the aggregation propensity of proteins prone to misfolding with SOD1(A4V) expression strongly suggest that the protein homeostasis system as a whole is dramatically altered by the introduction of a single disordered protein, conditions which are likely relevant to early stages of human pathology involving disordered protein expression.

## Materials and methods

### Plasmid construction

Constructs based on the pcDNA5 vector (Thermo Fisher Scientific, Waltham, MA) were used to express full-length HSPA1A and HSPA8 with N-terminal V5 and BirA epitope tags and a C-terminal in-frame K48R ubiquitin (pTP3923 and pTP3921, respectively), under the control of a doxycycline-inducible CMV promoter. pcDNA5/FRT/TO V5 HSPA8 was a gift from Harm Kampinga (Addgene plasmid #19514; http://n2t.net/addgene:19514; RRID:Addgene_ 19514, Watertown, MA) and pcDNA5/FRT/TO V5 HSPA1A was a gift from Harm Kampinga (Addgene plasmid #19510; http://n2t.net/addgene:19510; RRID:Addgene_19510, Watertown, MA) [88]. Versions containing deletions of the gly-gly amino acids at the end of ubiquitin were generated (pTP3924 and pTP3922, respectively), as well as versions containing ubiquitin only (pTP3939 and pTP3940). A lentivirus construct containing the biotin ligase BirA was generated from the lentiCRISPR v2 backbone (Addgene 52961, Watertown, MA) [89] and a construct containing BirA (a gift from Mauro Modesti) to make pTP3530, and used to make lentivirus with the helper plasmids pCMV-dR8.91 (Delta 8.9) and VSV-G. pLentiCRISPR v2 was a gift from Feng Zhang (Addgene plasmid #52961; http://n2t.net/addgene:52961; RRID: Addgene_52961, Watertown, MA). BioID2 constructs were made by inserting BioID2 (Addgene 92308, Watertown, MA) as an N-terminal fusion into HSPA8 and HSPA1A pcDNA5 constructs (Addgene 19514 and 19510, Watertown, MA) to make pTP4458 and pTP4459, respectively, retaining a V5 epitope tag in the linker. mycBioID2-13X Linker-MCS was a gift from Kyle Roux (Addgene plasmid # 92308; http://n2t.net/addgene:92308; RRID: Addgene_92308, Watertown, MA). Cloning details are available upon request. An MRE11-specific shRNA (5′-ACAGGAGAAGAGAUCAACUUUGuuaauauucauagCAAAGUUGAU CUCUUCUCCUGU-3′) was expressed under doxycycline control from pRSITEP-U6Tet-(sh)-EF1-TetRep-2A-Puro (Cellecta Mountain View, CA). BacMam constructs were generated from pAceBac1 (Geneva Biotech, Genève, Switzerland) with the Tet-On CMV promoter from pcDNA5 replacing the polyhedrin promoter. This construct was used to make bacmams expressing full-length MSH2 (pTP4651) and V5-tagged SOD1(A4V)-P2A-GFP (pTP4497), with MSH2 from MSH2/pBluescript [90] (Addgene #16453, Watertown, MA) and SOD1 from SOD1/pLX304 (DNASU #HsCD00440778, Tempe, AZ), respectively. The A4V mutation in SOD1 was made using QuikChange mutagenesis (Agilent, Santa Clara, CA). A version of the SOD1 construct with SOD1 removed (containing V5-tagged GFP only, pTP4552) was generated from pTP4497. These were used to generate bacmids and baculovirus according to manufacturer instructions for the bac-to-bac system (Thermo Fisher Scientific, Waltham, MA). ATP1B1 was overexpressed by transient transfection using a pcDNA3-based construct from GenScript (#OHu18298D, Piscataway, NJ). Plasmid sequences and cloning details are available upon request.

## Yeast genetics

A budding yeast strain lacking *SSA1*, *2*, *3*, and *4* [91] (MH272 3f MATalpha *ura3 leu2 ade2 his3* HMLa *rme1* GAL+ *ssa1*::TRP1 *ssa2 ssa3 ssa4*), a gift from Sabine Rospert containing a *URA3* plasmid with wild-type *SSA1*, was transformed with the *HIS3* plasmids pRS423 (vector), pRS423 containing wild-type *SSA1* (pTP4108), pRS423 containing wild-type *SSA1* with a C-terminal fusion of K48R human ubiquitin (pTP3988), or pRS423 containing wild-type *SSA1* with a C-terminal fusion of K48R human ubiquitin ΔGG (pTP3989). Yeast strains were streaked onto 5-FOA to select for loss of the *URA3* plasmid and reveal the complementation by the UBAIT constructs.

## Mammalian cell culture

We used a human osteosarcoma cell line (U2OS) containing an FRT recombination site integrated into the genome adjacent to a CMV promoter blocked by two copies of the Tet operator (Invitrogen Flp-In T-Rex system, Carlsbad, CA). HSPA1A, HSPA8, and ubiquitin UBAIT constructs were transfected into these cells and selected with hygromycin according to manufacturer's instructions. BirA-expressing lentivirus was then added and cells were selected with 1 μg/mL puromycin. Cells were then grown in Dulbecco's Modified Eagle Medium (DMEM) (Invitrogen, Carlsbad, CA) with 7.5% FBS (Tet-minus, Invitrogen, Carlsbad, CA) with 100 units/mL penicillin-streptomycin, 15 μg/mL blasticidin, 200 μg/mL hygromycin, and 1 μg/mL puromycin. Expression of UBAIT constructs was induced by addition of 1 μg/mL doxycycline the day after seeding in 15-cm dishes and harvested after 3 days. Expression of BioID2 constructs were induced by addition of 1 ug/mL doxycycline with 50 μM biotin supplement the day after seeding in 15-cm dishes and harvested after 3 days. Trypsinized cells were washed in PBS and flash-frozen using liquid nitrogen and kept at −80˚C until use.

## Lentivirus production and transduction

HEK-293T cells (ATCC) were grown in DMEM with 7.5% FBS. The 293T cells were plated in 6-well dishes and allowed to grow to near confluence. A solution with optiMEM and plasmids was made using 1.2 μg of pLX304 vector containing shRNA for the gene of interest, 100 ng of VSV-G, 500 ng of pCMV-dR8.91 (Δ 8.9), and brought to 100 μL with Opti-MEM. Seven microliters of FuGENE (Promega, Madison, WI) was combined with the Opti-MEM solution, incubated for 25 minutes at room temperature, and then added to the 293T cells in 6 wells. The media was changed a day later and left for 24 hours. The media was harvested, then replaced, and again harvested 24 hours later. The collected media was combined and filter sterilized using 0.45-μm filters and then stored at −80˚C until needed. U2OS cells were plated in 6-well plates and allowed to reach near confluency. The media was removed and 1 mL of the viral aliquots was added. The cells were grown overnight and then media was replaced 24 hours later. The following day, cells were transferred to a T-75 flask with selection agent added. Cells were incubated for approximately 1 week until control cells without virus died. For transient transfection of the ATP1B1 expression plasmid, U2OS cells were plated in a 15-cm dish prior to addition of doxycycline for expression of UBAIT constructs. A total of 12 μg of plasmid was added with calcium phosphate per 15-cm dish, as previously described [92].

## Streptavidin isolation of biotin-tagged UBAIT targets and BioID targets

Each U2OS cell pellet (from four 15-cm dishes) was lysed in 1 mL of lysis buffer (8 M urea, 50 mM Tris pH 8, 5 mM CaCl$_2$, 500 mM NaCl, 0.1% SDS, 2.5 mM PMSF [Calbiochem, San

Diego, CA], and EDTA-free Protease Inhibitor [Pierce, Appleton, WI]). Lysates were sonicated twice for 30 seconds, separated by a 30-second rest. A volume of lysate containing 3 mg protein was brought to a final volume of 1 mL in lysis buffer; urea concentration was then reduced to 1 M with dilution buffer (50 mM Tris pH 8, 500 mM NaCl, 0.1% SDS) by gradual addition while vortexing. At least two samples of 3 mg each were typically obtained from each pellet. M-280 Streptavidin Dynabeads (Invitrogen, 11206D, Carlsbad, CA) were washed twice in dilution buffer, resuspended in the same volume, and 120 μL of bead resuspension was added to each sample and samples were rotated overnight at room temperature. Beads were then resuspended in 1.5 mL wash buffer (1 M urea, 50 mM Tris pH 8, 500 mM NaCl, 0.1% SDS). Buffer was removed and beads were then washed twice for >30 minutes in wash buffer, followed by 500 mM LiCl for 15 minutes, then 0.1% SDS, 0.2% SDS, and 0.5% SDS for 30 minutes each. All wash steps were performed in 1.5 mL of solution while rotating at room temperature. Bound protein was eluted by boiling beads at 100°C for 5 minutes in 50 μL of 1% SDS solution containing 50 mM 2-mercaptoethanol. The elution step was repeated and eluates were combined and stored at −20°C.

## Filter aided sample preparation, mass spectrometry, and quantification

Frozen eluates were boiled for 2 minutes at 100°C, then any residual beads were removed with a magnet. Samples were diluted with 600 uL of UA buffer (8 M urea, 0.1 M Tris pH 8.8). MicroCon-30 centrifugal filter units (Millipore, MRCF0R030, Burlington, MA) were equilibrated with 20% ACN, 2% formic acid solution, and centrifuged at 14,000*g* for 10 minutes prior to use. Samples were loaded onto the filters then washed 3 times with 400 μL of UA buffer. After washing, samples were incubated for 5 minutes at room temperature with 400 μL of 50 mM DTT in UA buffer to reduce disulfide bonds. Samples were centrifuged and then alkylated with 400 μL of 50 mM iodoacetamide by incubation for 5 minutes at room temperature, followed by centrifugation. Samples were desalted with 400 μL of 40 mM ammonium bicarbonate (ABC) 3 times. A total of 100 μL of 40 mM ABC containing 0.5 μL of trypsin gold (Promega, V528A, Madison, WI) in PBS was added to each sample, and samples were incubated overnight at 37°C in a closed, humidified chamber. Peptides were eluted by centrifugation and filtrate was reserved in tube; filters were then washed with 100 uL 20% ACN, 2% formic acid solution, and filtrate was combined with eluted peptides in ABC buffer. Collected samples were lyophilized at room temperature. Dried samples were resuspended in 10 μL 0.1% formic acid with 0.1% trifluoroacetic acid, then desalted with C18 tips (Pierce, QK224796, Appleton, WI) according to the manufacturer's protocol. The final samples were resuspended in 80% ACN 2% formic acid for LC-MS analysis. All centrifugations were done at 14,000*g* for 10 minutes at room temperature unless otherwise noted. Protein identification by LC-MS/MS was provided by the University of Texas at Austin Proteomics Facility on an Orbitrap Fusion following previously published procedures [84]. Raw files were analyzed using label-free quantification with Proteome Discoverer 2.2. The details of the workflow for PD 2.2 are available upon request.

## Statistics testing and analysis

Proteome Discover 2.2 results were further refined by two additional methods in order to control the FDR. First, all proteins were cross-referenced for common contaminants, in which case they were removed from final analysis. Any polypeptides with fewer than two unique peptides identified were removed from the final analysis. For UBAIT samples, refined data were then normalized by levels of total HSP70 or HSC70 in order to correct for variation of recovery between samples. For BioID samples, refined data were normalized by total peptide counts.

Missing data were imputed using weighted low abundance resampling, which replaces missing values with random values sampled from the lower 5% of the detected values, with heavier weighting toward higher values.

To compare negative control ΔGG UBAIT with experimental wild-type UBAIT samples, we took the ratio of average intensity measurements from wild-type UBAITS divided by ΔGG UBAITS for each protein identified to create an enrichment ratio. A bootstrapping method was then used to create a distribution of possible ratios (1,000) for each protein by randomly assigning wild-type or ΔGG values from each protein. Finally, we compared the actual ratio to the hypothetical distribution of randomized ratios and computed the *p*-value of the actual ratio from the quantile. Proteins with ratios within the top 5% percentile were considered to be significant. In order to control the FDR from multiple hypothesis testing, we used a modified Benjamini-Hochberg procedure [27] for each of the experiments using an FDR of 0.05 (see S1 Data). In this procedure, we used the lowest critical-value that was greater than the *p*-value to maximize estimation of potential false positives.

A similar approach was used for BioID; instead of using ΔGG UBAIT data as the negative control, BioID only (no chaperone) samples were used. Bootstrapping was performed as described above to compare HSP70-BioID/HSC70-BioID and BioID only.

### SILAC

U2OS cells containing HSPA1A and HSPA8 UBAITs were grown in Light Lysine/Arginine DMEM with 10% dialyzed FBS for at least 2 weeks (at least 5 cell doublings). U2OS cells were plated in 15-cm dishes with 20 mL of Light DMEM. The next day, media was replaced by Heavy Lysine ($^{13}C_6$, $^{15}N_2$ L-Lysine-2HCl), Arginine ($^{13}C_6$, $^{15}N_4$ L-Arginine-HCl) DMEM (Thermo Fisher Scientific, Waltham, MA) with 10% dialyzed FBS containing 1 ug/mL doxycycline and grown for either 1 or 2 days. Cells were harvested and frozen as described above. Cell pellets were lysed and UBAIT targets were isolated as described above. Twelve samples were used for each chaperone (wild-type UBAIT only). These samples, as well as 3 total lysates, were analyzed by mass spectrometry for protein identification as well as light/heavy ratios for each peptide. Total cell lysate light/heavy ratio data were used to experimentally derive two data points for each enriched protein identified (day 1 and day 2). Using day 0 as 100% light, three points were used to calculate a linear slope explaining the change of the light/heavy ratio for each individual protein. The averaged slope from the 3 points was used to calculate expected light/heavy ratios for UBAIT targets (the "expected" values, see S4 Data). Light/heavy ratios were also measured from 12 wild-type UBAIT isolations from day 1 and day 2 SILAC experiments (the "actual" values, S4 Data). No ΔGG isolations were performed in this case, but proteins positively identified as HSC70 and HSP70 binding partners in experiments described in Fig 1 were compared with the expected value using a one-sample *t* test.

### Isolation of detergent-resistant aggregates

This procedure was performed as previously described [84].

### Supporting information

**S1 Fig. Levels of HSC70 and HSP70 UBAIT are low in comparison to endogenous HSC70.** (A) Western blot of HSC70 UBAITs expressed in human U2OS cells treated with doxycycline (Dox) (1 ug/mL) for 3 days or untreated, using anti-HSC70 (Santa Cruz sc-7298). (B) Western blot of HSP70/HSC70 UBAITs expressed in human U2OS cells treated with Dox (1 ug/mL) for 3 days or untreated, using streptavidin-AlexaFluor680 (Life Technologies). (C) U2OS cells and U2OS cells expressing the ubiquitin-tagged UBAIT constructs (with doxycycline) were

analyzed by immunofluorescence using antibodies directed against HSP70 (Enzo ADI-SPA-810) or HSC70 (Santa Cruz sc-7298) and imaged by confocal microscopy, with DAPI as the counterstain, as indicated. HSC, heat shock cognate; HSP, heat shock protein; UBAIT, ubiquitin-activated interaction trap.
(PDF)

**S2 Fig. ΔGG UBAIT *SSA1* complements *S. cerevisiae* deficient in HSP70 chaperones, but wild-type UBAIT *SSA1* does not.** An *S. cerevisiae* strain deficient in *SSA1*, *SSA2*, *SSA3*, and *SSA4* was complemented by vector only, wild-type *SSA1*, *SSA1* UBAIT (C-terminal ubiquitin fusion), or *SSA1* UBAIT ΔGG (C-terminal ubiquitin fusion lacking GG at the C terminus), as indicated. Strains were streaked onto 5-FOA media, which selects for loss of the URA3 (wt *SSA1*) plasmid, maintaining viability of the *ssa1-4* strain. See also S6 Data. *SSA1*, stress-seventy subfamily A 1; UBAIT, ubiquitin-activated interaction trap; wt, wild type; 5-FOA, 5-Fluor-oorotic acid.
(PDF)

**S3 Fig. Examples of target enrichment in HSP70 and HSC70 UBAIT isolations.** (A, B, C) Levels of binding of specific targets to HSC70 UBAIT wild-type or ΔGG isolations (normalized by level of HSC70 expression), with 12 replicates shown per sample. (D, E, F) Levels of binding of specific targets to HSP70 UBAIT wild-type or ΔGG isolations (normalized by level of HSP70 expression). All examples shown yield enrichment values that exceed the 95% confidence interval and are retained at FDR 0.05 using Benjamini-Hochberg (see Materials and methods for details). FDR, false discovery rate; HSC, heat shock cognate; HSP, heat shock protein; UBAIT, ubiquitin-activated interaction trap.
(PDF)

**S4 Fig. Functions of enriched targets identified from the HSC70 and HSP70 UBAIT experiments (only targets found to be significant in all 3 sets).** Fold enrichment values are shown for the top gene ontology categories among the HSC70 UBAIT targets (top panel) and the HSP70 UBAIT targets (bottom panel). FDR, $p < 0.05$ only. FDR, false discovery rate; HSC, heat shock cognate; HSP, heat shock protein; UBAIT, ubiquitin-activated interaction trap.
(PDF)

**S5 Fig. Significant targets identified from HSP70 or HSC70 UBAIT isolations containing either wild-type ubiquitin or K48R ubiquitin (K48R Ub: $N = 3$ groups, 12 samples wild-type and 12 samples ΔGG each; WT Ub: $N = 2$ groups, 12 samples WT and 12 samples ΔGG each).** Error bars show standard deviation. See also S6 Data. HSC, heat shock cognate; HSP, heat shock protein; UBAIT, ubiquitin-activated interaction trap; WT, wild-type.
(PDF)

**S6 Fig. Significant targets identified from HSC70 UBAIT isolations containing either wild-type or V438F HSC70 ubiquitin.** (A) Venn diagrams of wild-type and V438F HSC70 UBAIT targets identified, each with $N = 6$, all with K48R ubiquitin fusions. (B) Western blot of HSC70 UBAITs expressed in human U2OS cells treated with doxycycline (Dox) (1 ug/mL) for 3 days or untreated, using streptavidin-AlexaFluor680 (Life Technologies). (C) Summary of average WALTZ [47] and TANGO [46] scores of significant targets, as well as polypeptide length of proteins enriched with UBAITs in cells expressing wild-type or V438F HSC70. Top: analysis including shared targets (239 WT versus 251 VF); bottom: analysis excluding shared targets (111 WT versus 123 VF). Welch's one-tailed *t* test was used to compute *p*-values. HSC, heat shock cognate; UBAIT, ubiquitin-activated interaction trap; VF, V438F; WT, wild type.
(PDF)

**S7 Fig. BioID identification of HSP70 binding partners.** (A) Number of HSP70 binding partners identified by BioID2 (12 replicates [1 set]) compared to HSP70 UBAIT (significant at FDR 0.05 in at least one set). (B) Western blot of inducible BioID2 and BioID2-HSP70 fusion protein expression in U2OS cells, with biotin and doxycyline addition as indicated; visualized with streptavidin-AlexaFluor680 (Life Technologies). (C) Schematic diagram of targets identified through BioID2-HSP70 and UBAIT HSP70. Targets found in one or two sets (but not all three). UBAIT experiments are shown with a dashed line. (D, E, F) Levels of binding of specific targets to BioID2-HSP70 or BioID2 alone. All examples shown yield enrichment values that exceed the 95% confidence interval and are retained at FDR 0.05 using Benjamini-Hochberg (see Materials and methods for details). See also S6 Data. FDR, false discovery rate; HSP, heat shock protein; UBAIT, ubiquitin-activated interaction trap.
(PDF)

**S1 Data. Mass spectrometry data from HSC70, HSP70, and Ub-only UBAIT experiments (Fig 1).** HSC, heat shock cognate; HSP, heat shock protein; Ub, ubiquitin; UBAIT, ubiquitin-activated interaction trap.
(XLSX)

**S2 Data. Mass spectrometry data from HSC70 V438F UBAIT experiment (S6 Fig).** HSC, heat shock cognate; UBAIT, ubiquitin-activated interaction trap.
(XLSX)

**S3 Data. Mass spectrometry data from HSP70 BioID experiment (S7 Fig).** HSP, heat shock protein.
(XLSX)

**S4 Data. Mass spectrometry data from HSC70 and HSP70 SILAC UBAIT experiment (Fig 2).** HSC, heat shock cognate; HSP, heat shock protein; SILAC, stable isotope labeling with amino acids in cell culture; UBAIT, ubiquitin-activated interaction trap.
(XLSX)

**S5 Data. Mass spectrometry data from HSC70 and HSP70 UBAiT experiment with SOD1 expression (Fig 4).** HSC, heat shock cognate; HSP, heat shock protein; SOD1, superoxide dismutase 1; UBAIT, ubiquitin-activated interaction trap.
(XLSX)

**S6 Data. Supporting data values for Figs 2, 3, 4, S3, S5 and S7.**
(XLSX)

## Acknowledgments

We thank the CBRS Proteomics Core Facility and Dr. Maria Person for assistance with mass spectrometry, Xuemei Wen for molecular biology assistance, and members of the Paull lab for helpful comments on the manuscript. We also thank Sabine Rospert for the *Δssa1-4* yeast strain [91], Harm Kampinga for the HSPA8 and HSPA1A clones (Addgene plasmids #19514 and 19510) [88], Bert Vogelstein for the hMSH2 Clone E3 (Addgene plasmid #16453) [90], Feng Zhang for the lentiCRISPR construct (Addgene plasmid #52961) [89], Kyle Roux for the BioID2 construct (Addgene plasmid #92308) [70], and Mauro Modesti for the BirA plasmid. This work was supported in part by the Howard Hughes Medical Institute.

## Author Contributions

**Conceptualization:** Seung W. Ryu, Jon M. Huibregtse, Tanya T. Paull.

**Data curation:** Seung W. Ryu, Nicolette A. Ender, Antony Harvey, Tanya T. Paull.

**Formal analysis:** Seung W. Ryu, Carlos P. Zanini, Antony Harvey, Peter Mueller, Tanya T. Paull.

**Funding acquisition:** Tanya T. Paull.

**Investigation:** Seung W. Ryu, Rose Stewart, D. Chase Pectol, Nicolette A. Ender, Oshadi Wimalarathne, Ji-Hoon Lee, Antony Harvey, Tanya T. Paull.

**Methodology:** Seung W. Ryu, Rose Stewart, D. Chase Pectol, Nicolette A. Ender, Oshadi Wimalarathne, Ji-Hoon Lee, Antony Harvey, Tanya T. Paull.

**Project administration:** Seung W. Ryu, Tanya T. Paull.

**Resources:** Tanya T. Paull.

**Software:** Seung W. Ryu, Carlos P. Zanini, Antony Harvey.

**Supervision:** Tanya T. Paull.

**Validation:** Seung W. Ryu, Carlos P. Zanini, Jon M. Huibregtse, Peter Mueller.

**Visualization:** Seung W. Ryu, Tanya T. Paull.

**Writing – original draft:** Seung W. Ryu, Tanya T. Paull.

**Writing – review & editing:** Jon M. Huibregtse, Peter Mueller, Tanya T. Paull.

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
