## [Editor Report · Decision Letter 0]

27 Nov 2019

Dear Tanya, 

Thank you for submitting the revised version of your manuscript entitled "Comprehensive identification of HSP70/HSC70 Chaperone Clients in Human Cells" for consideration as a Methods and Resources by PLOS Biology.

Your manuscript has now been evaluated by the PLOS Biology editorial staff as well as by an academic editor with relevant expertise and I am writing to let you know that we would like to send your submission out for external peer review.

Please re-submit your manuscript within two working days, i.e. by Nov 29 2019 11:59PM.

Kind regards,

Ines

--

Ines Alvarez-Garcia, PhD

Senior Editor

PLOS Biology

Carlyle House, Carlyle Road

Cambridge, CB4 3DN

+44 1223–442810

---

## [Decision Letter · Decision Letter 1]

20 Dec 2019

Dear Tanya,

Thank you very much for submitting your manuscript "Comprehensive identification of HSP70/HSC70 Chaperone Clients in Human Cells" for consideration as a Methods and Resources at PLOS Biology. Your manuscript has been evaluated by the PLOS Biology editors, an Academic Editor with relevant expertise, and by three independent reviewers.

As you will see, the reviewers find the use of the method to identify HSP70 and HSC70 chaperone clients interesting and novel, and the identification of potential candidates significant for the field. However, they also raise several issues that need to be addressed before we can consider the paper for publication. The main issues are: 1) using a different method to validate the candidates and 2) the inclusion of a control experiment using HSP70/HSC70 defective for substrate binding. In addition, you should address the concern raised by Reviewer 1 that the UBAIT technique is not appropriate for non-E3s by including further validation and performing the suggested controls.

The reviews of your manuscript are appended below. Based on their specific comments and following discussion with the academic editor, I regret that we cannot accept the current version of the manuscript for publication. However, we remain interested in your study and we would be willing to consider resubmission of a comprehensively revised version that thoroughly addresses all the reviewers' comments. We cannot make any decision about publication until we have seen the revised manuscript and your response to the reviewers' comments. Your revised manuscript would be sent for further evaluation by the reviewers.

We appreciate that these requests represent a great deal of extra work, and we are willing to relax our standard revision time to allow you six months to revise your manuscript.We expect to receive your revised manuscript within 6 months.

**IMPORTANT - SUBMITTING YOUR REVISION**

*NOTE: In your point by point response to to the reviewers, please provide the full context of each review. Do not selectively quote paragraphs or sentences to reply to. The entire set of reviewer comments should be present in full and each specific point should be responded to individually, point by point.

*Resubmission Checklist*

*Published Peer Review*

*PLOS Data Policy*

*Blot and Gel Data Policy*

Best wishes,

Ines

--

Ines Alvarez-Garcia, PhD

Senior Editor

PLOS Biology

Carlyle House, Carlyle Road

Cambridge, CB4 3DN

+44 1223–442810

Reviewers’ comments

Rev. 1:

In this paper, Ryu et al profiled the clients of HSP70/HSC70 in human cells. They used a method named UBAIT, previously developed to capture interacting proteins of ubiquitin ligases, to identify HSP70/HSC70 interacting proteins. They show that HSP70/HSC70 have different client preference and a single misfolded protein SOD1 (A4V) induces changes in HSP70/HSC70 client association.

The major caveat in this study is that UBAIT was originally designed to study ubiquitin ligases (E3s) because E3s contain either a HECT or a RING domain that is necessary for ubiquitin transfer from E2 to substrates. However, HSC70 does not contain any characteristic E3 domains, thus when HSC70-Ub forms thioester with E2, it is unlikely to label HSC70 target directly. Instead, it will function with other E3s in cells, and potentially label other ubiquitin targets. It is possible that HSC70 and its targets are also labeled but the overall background will be significantly high. The cellular ubiquitination machineries will just take HSC70-Ub as a regular ubiquitin and incorporate HSC70-Ub into other ubiquitinated proteins in cells. The deltaGG control experiment cannot exclude this possibility since it is an inactive ubiquitin. The authors did profile all ubiquitinated proteins in cells using biotin-V5-ubiquitin only and claimed there are not much overlap between biotin-V5-ubiquitin and biotin-V5-HSC70-ubiquitin. However, only 52 unique proteins were identified with biotin-V5-ubiquitin. This is contradictory to many previous studies. Mass spectrometry can easily identify thousands of ubiquitinated proteins with tagged ubiquitin. Overall, the conclusions made in this paper seems all based on a method that is not validated to be useful for non-E3 proteins. The conclusions can alternatively be explained by the altered ubiquitination status of the “client” proteins. Here are other major concerns:

(1) The discovery and validation experiments are all based on UBAIT. First, the authors need to establish the proof of principle why this method can be used for non-E3s. Is there any covalent modification of known HSP70 client? How is the background? The current Biotin-V5-ubiquitin dataset is not convincing for the reason stated above. Secondly, the validation of the binding partners (Fig 3) should not use UBAIT since it is the method used to identify these potential binding partners. Alternative validation methods, such as co-IP, should be used for validation.

(2) Instead of deltaGG, a potential good control will be a HSP70 mutant that is defective in substrate binding.

(3) It is generally considered that chaperons assist protein folding, so it is not surprising that HSP70 partners are enriched for newly synthesized proteins. It is not obvious to the reader what proteins are regulated by HSP70 or HSC70 unless one checks the supplementary tables. It will be more informative to extend the study in Fig 2 to include more discussion about what are those proteins. Are those proteins enriched in particular biological pathway? Any functional relevance?

(4) The paper also lacks the general information about what are the new HSP70 clients that are not previously identified.

Minor concerns:

(1) Fig 1B, the molecular weight marker should be added.

(2) Fig 4E, is the y-axis the same as in Fig 3A? Why so different?

(3) Duplications in the reference. Reference paper 25 is the same as 28.

Rev. 2:

Ryo et al compare the substrate pool of human Hsp70 and Hsc70. The authors address a very important question, to which extent do Hsp70 homologues differ in substrate selectivity and hence in function. If such a case is convincingly made, it would be interesting for a broad readership. The authors identify substrates employing C-terminal ubiquitin fusions to subsequently create covalent substrate complexes that can be identified by mass spectrometry. The authors identify differences between Hsp70 and Hsc70. However, there are some serious concern that question the relevance of the finding and preclude publication in the present form.

Major concerns

1. The top hits identified by the authors contain highly positively charged proteins, many of which are nucleotide binding proteins. Others are proteins interacting with the ubiquitination machinery. As Hsp70s are strongly negatively charged, the identification of these substrates may be caused by coulomb interactions. This is even more likely as they work in overproduction conditions, favouring aberrant binding.

2. The authors need to include Hsp70 mutants allowing to comment on functional and specific interactions. This would be the classic mutations blocking ATP hydrolysis in the ATPase domain and the Val to Phe mutations blocking substrate binding via the substrate binding pocket.

3. The substrate data sets are poorly presented and analysed. The authors need to analyse to which protein classes their substrates belong to, and to which extent they identify substrates previously identified, and what would be common features (e.g. function, disorder, charge etc.).

4. The C-terminal fusions preclude interaction with EEVD binding co-chaperones. It is unclear how and why the authors could identify some of the co-chaperones that bind to Hsp70 via this motif, e.g. Hop.

5. The authors claim the mutants are functional as Ub fusions complement in yeast. Such data need to be provided. They should be discussed keeping in mind that not all functions of Hsp70 chaperones under permissive conditions are essential functions

Rev. 3:

In this study, the authors employed a ubiquitin-mediated proximity ligation strategy (UBAIT) to covalently trap the binding partners of the human molecular chaperones HSC70 and HSP70. This system was originally utilized to map the interactors of ubiquitin ligases, but the K48R mutation makes it a versatile tool for other proteins of interest also. Like BioID and APEX, this approach can better capture direct, transient interactors compared to native affinity purification methods. Despite the wealth of biochemical data on the Hsp70 family of chaperones, their endogenous substrates have not been systematically mapped. Ribosome profiling has been used to globally map co-translational Hsp70 clients in yeast, but no proteomic datasets were previously generated. Since all Hsp70 homologs that have been studied in vitro preferentially bind to hydrophobic peptide sequences, it is generally assumed that all paralogs in a given organism would have indistinguishable client repertoires. However, by fusing human HSC70 and HSP70 to ubiquitin (K48R), expressing these constructs in HeLa cells, affinity purifying the chaperone-client conjugates and performing mass spectrometry, the authors found that HSC70 and HSP70 have a large set of non-overlapping clients. As expected based on the fact that in mature proteins hydrophobic binding sites are usually buried in protein cores, interaction interfaces and membranes, both HSC70 and HSP70 preferentially associate with nascent proteins and protein complex members lacking interaction partners. Finally, they show that that expression of an intrinsically misfolded protein (an ALS-associated SOD1 mutant) alters the landscape of HSC70 and HSP70 binding partners.

Comments

1) Overall the study is novel and provides a valuable resource that had been conspicuously absent in the literature.

2) The UBAIT strategy is a clever way to capture chaperone-client interactions.

3) The paper is well-written and the logic is easy to follow.

4) The ubiquitin fusion proteins are ectopically expressed on top of the wild type versions. The authors should assess what fraction of the total HSC70 and HSP70 is the UBAIT fusion.

5) The authors should show the data demonstrating that HSC70-UBAIT is functional (complements the yeast deletion). The conserved "EEVD" sequence at the C-terminus is thought to be untaggable, so this should be commented on.

6) HSP70 (HSPA1A) is not typically expressed under basal conditions, so the authors should caveat that the HSP70 results are not physiological.

7) The authors should comment on the identification of ER resident proteins (e.g., DNAJC proteins). Is this occurring post-lysis, or is the protein getting into the ER?

8) It would be reassuring to show by IF that the UBAIT tagging does not alter the subcellular localization of HSC70/HSP70.

9) The word "comprehensive" in the title is too strong and impossible to verify. Moreover, multiple cell lines were not examined. "Proteome-wide identification of HSP70/HSC70 chaperone clients in a human cell line" would be more accurate.

10) The comparison of the BioID vs UBAIT strategies is imperfect since the author utilized different termini of HSP70 (C terminal for UBAIT and N terminal for BioID2) for tagging. Since the BioID tag is on the N-terminus connected to the ATPase domain, it is more likely to get all the cochaperones (evidenced by DNAJB1), but farther from the substrate binding domain at the C-terminus. This should be caveated.

11) Also, a detailed method for the BioID approach is lacking.

Minor Comments

1) Labelling:

a) label is missing in the bar graph of Figure 3C.

b) In all Figures for consistency label use either dGG or ∆GG.

c) Figure 4B, be consistent with HSP70 UBAIT or HSP70 Ubait

---

## [Decision Letter · Decision Letter 2]

18 May 2020

Dear Tanya,

Thank you for submitting your revised Methods and Resources entitled "Proteome-wide identification of HSP70/HSC70 Chaperone Clients in Human Cells" for publication in PLOS Biology. Thank you also for your patience as we completed our editorial process, and please accept my apologies for the delay in providing you with our decision. I have now obtained advice from the three original reviewers and have discussed their comments with the Academic Editor. 

We're delighted to let you know that we're now editorially satisfied with your manuscript. You will see that two of the reviewers are now completely satisfied, whereas Reviewer 2 still raises some issues. After discussing these points with the academic editor, we find your responses compelling and feel that the V438F mutant provides new interesting information, adds to the model and has consistencies with previous findings in the literature.

Before we can formally accept your paper and consider it "in press", we also need to ensure that your article conforms to our guidelines. A member of our team will be in touch shortly with a set of requests. As we can't proceed until these requirements are met, your swift response will help prevent delays to publication. Please also make sure to address the data and other policy-related requests noted at the end of this email.

*Copyediting*

*Published Peer Review History*

*Early Version*

*Submitting Your Revision*

Best wishes,

Ines

--

Ines Alvarez-Garcia, PhD

Senior Editor

PLOS Biology

Carlyle House, Carlyle Road

Cambridge, CB4 3DN

+44 1223–442810

Fig. 2F; Fig. 3A, B, C; Fig. 4A, E; Fig. S3A, B, C, D, E, F; Fig. S5 and Fig. S7D, E, F

Reviewers’ comments

Rev. 1:

The authors have fully addressed all the concerns.

Rev. 2:

Ryu et al provide a revised version, in which the key addition is a V to F mutation in the substrate binding cleft which is known to have dramatically reduced substrate binding affinity. Such a mutant should be useful to separate relevant from irrelevant substrates. However, this mutant attracts as many unique hits compared to the wild type as the other way round. This makes it unlikely that this method is appropriate to identify specific interactors of Hsp70 chaperones.

The main concerns are:

1. The V to F mutant attracts 123 unique substrates, compared to the 111 unique substrates of the wildtype. Looking at the diverse of substrates of the V to F mutant, there is no pattern that can explain meaningful binding to this mutant en mass. It demonstrates that this method unfortunately attracts many unspecific binders, and in this case it is more than the total binding of the wildtype protein.

2. The data presentation is not acceptable for publication. The authors argue about properties of the identified substrate pools regarding sequence properties such as charges or prediction scores, but they do not show the data anywhere. From the prediction data only the averages are shown. This does not take away the suspicion that the top hits are driven by charge. To make the statements in the point-by-point reply the authors apparently have done this for every single protein. There is no excuse not to add this information to the protein hits in the supplementary tables.

3. The authors had been asked to use an ATPase deficient mutant as control. They completely ignore this request. Such as mutant, such as the known Y to A mutation in the ATPase domain, would allow to separate functionally loaded proteins, i.e. such that enter the Hsp70 chaperone properly via the ATPase cycle, from unspecific binders. It is surprising that the authors did not realise that this would be a different answer than the one with the V to F mutation in the substrate binding pocket.

Rev. 3:

The authors have thoroughly addressed the reviewer comments, added important new data and significantly improved the paper. I have no further comments or concerns.

---

## [Editor Report · Decision Letter 3]

29 Jun 2020

Dear Dr Paull,

On behalf of my colleagues and the Academic Editor, Kylie J. Walters, I am pleased to inform you that we will be delighted to publish your Methods and Resources in PLOS Biology. 

Early Version

PRESS 

Kind regards,

Pamela Berkman

Publishing Editor

PLOS Biology

on behalf of

Ines Alvarez-Garcia,

Senior Editor

PLOS Biology